# The Long Non-Coding RNA *GOMAFU* in Schizophrenia: Function, Disease Risk, and Beyond

**DOI:** 10.3390/cells11121949

**Published:** 2022-06-17

**Authors:** Paul M. Zakutansky, Yue Feng

**Affiliations:** 1Graduate Program in Biochemistry, Cell and Developmental Biology, Emory University, Atlanta, GA 30322, USA; pzakuta@emory.edu; 2Department of Pharmacology and Chemical Biology, Emory University School of Medicine, Atlanta, GA 30322, USA

**Keywords:** long non-coding RNA, schizophrenia, genetic risk, *GOMAFU*, RNA-binding proteins

## Abstract

Neuropsychiatric diseases are among the most common brain developmental disorders, represented by schizophrenia (SZ). The complex multifactorial etiology of SZ remains poorly understood, which reflects genetic vulnerabilities and environmental risks that affect numerous genes and biological pathways. Besides the dysregulation of protein-coding genes, recent discoveries demonstrate that abnormalities associated with non-coding RNAs, including microRNAs and long non-coding RNAs (lncRNAs), also contribute to the pathogenesis of SZ. lncRNAs are an actively evolving family of non-coding RNAs that harbor greater than 200 nucleotides but do not encode for proteins. In general, lncRNA genes are poorly conserved. The large number of lncRNAs specifically expressed in the human brain, together with the genetic alterations and dysregulation of lncRNA genes in the SZ brain, suggests a critical role in normal cognitive function and the pathogenesis of neuropsychiatric diseases. A particular lncRNA of interest is *GOMAFU*, also known as *MIAT* and *RNCR2*. Growing evidence suggests the function of *GOMAFU* in governing neuronal development and its potential roles as a risk factor and biomarker for SZ, which will be reviewed in this article. Moreover, we discuss the potential mechanisms through which *GOMAFU* regulates molecular pathways, including its subcellular localization and interaction with RNA-binding proteins, and how interruption to *GOMAFU* pathways may contribute to the pathogenesis of SZ.

## 1. Introduction

Greater than 98% of the human genome contains non-coding sequences, which is largely transcribed to produce numerous non-coding RNAs, which include microRNAs and long non-coding RNAs (lncRNAs) [1,2]. lncRNAs, which harbor more than 200 nucleotides in length but do not encode proteins, have attracted increasing attention due to their function in normal brain development and their potential involvement in neuropsychiatric diseases. lncRNAs play multifaceted roles in gene regulation through nuclear domain organization, chromatin modulation, transcriptional regulation, governing RNA-binding protein localization and activity, as well as the regulation of mRNA translation and stability [3,4,5,6,7]. Despite discoveries of genetic mutations and the abnormal expression of lncRNAs in various mental illnesses [8,9,10,11], the connection of lncRNA mutation and dysregulation to the downstream molecular network underlying the pathophysiology of brain diseases is a major knowledge gap.

In this review, we focus on the lncRNA *GOMAFU*, a postulated risk factor for schizophrenia (SZ). *GOMAFU* was initially identified in mouse retinal cells as *retinal non-coding RNA 2 (Rncr2)* as well as in myocardial infarction patients, termed *myocardial infarction associated transcript (MIAT)*. However, growing evidence supports the function of this lncRNA in normal neuronal development and plasticity. Moreover, *GOMAFU* is dysregulated in postmortem brain neurons of SZ patients and potentially contributes to the pathogenesis of SZ. We review recent evidence as to how genetic alterations in the human *GOMAFU* gene distinctly correlate with SZ and myocardial infarction. In addition, we highlight the neurophysiological functions of *GOMAFU*, abnormal *GOMAFU* expression in SZ brains, and the potential usage of *GOMAFU* as a biomarker for SZ diagnosis and treatment. We further elaborate on the interactions of *GOMAFU* with RNA-binding proteins (RBPs) and possible regulatory loops on RNA homeostasis by the *GOMAFU*-RBP interactome.

## 2. Schizophrenia, Candidate Genes, and Dysregulated Alternative Splicing during Neurodevelopment

Schizophrenia (SZ), characterized by paranoia, cognitive impairment, and delusions, is known to affect 1% of the population of the world [12,13,14,15,16]. Moreover, SZ is shown to progress in a chronic manner and is linked to high morbidity and mortality rates [17,18]. The complex multifactorial etiology of SZ, which integrates genetic alterations, biological dysfunctions, and environmental factors, is poorly understood [19,20]. Proper neuronal development and neural circuitry plasticity is essential for normal brain function [21]. Abnormalities in neuronal development are clearly indicated in the pathogenesis of SZ [22,23,24,25,26]. In addition, glial abnormalities have also been found in SZ brains that disrupt functional interactions between glia and neurons, including dysregulation in oligodendrocyte-specific genes, resulting in subsequent alterations in oligodendrocyte development and myelination in SZ patients [27,28,29]. Consistent with the complex etiology of SZ, several transcriptome studies have identified numerous dysregulated genes in SZ brains [30,31,32]. To determine factors contributing to gene dysregulation, genome-wide association studies (GWAS) and linkage analysis of candidate genes have identified countless single nucleotide polymorphisms (SNPs) in neurodevelopmental genes that are associated with SZ [26,33,34,35,36]. Moreover, the fact that numerous genetic variants associated with SZ are found in introns further suggests the potential contribution of aberrant alternative splicing of pre-mRNA to SZ etiology and susceptibility [37,38,39,40,41].

Alternative splicing often generates two or more protein isoforms from a single gene, which has become a recognizable mechanism of gene dysregulation [42,43]. Strikingly, more than 40% of the transcripts expressed in the human brain are alternatively spliced [44], contributing to the complexity of gene regulation in the brain [30,45]. A recent consortium study documented genome-wide dysregulation of alternative splicing in SZ brains [30]. Moreover, dysregulated alternative splicing of the SZ risk gene transcripts that give rise to isoforms of the v-erb-a erythroblastic leukemia viral oncogene homolog 4 *(ErbB4)*, a receptor for another SZ candidate gene, Neuregulin 1 *(NRG1)*, have been well documented [38,46,47,48]. *ErbB4* produces four distinct isoforms which differ within either their extracellular juxtamembrane domain (JM), *JM-a* and *JM-b*, or their cytoplasmic tail region (CYT), *CYT-1*, and *CYT-2* [38,47,49]. Two independent studies reported significantly increased *JM-a* and *CYT-1* isoforms in the dorsolateral prefrontal cortex of SZ patients compared to controls [38,47]. Interestingly, SNPs located in the *ErbB4* gene were selectively associated with the distinct *ErbB4* isoforms dysregulated in SZ [38,46]. These data provide foundational evidence for the linkage of alternatively spliced isoforms to disease susceptibility.

An additional SZ candidate gene that has been extensively analyzed is Disrupted-In-Schizophrenia 1 (*DISC1*). In an initial study of a Scottish family, a balanced (1;11)(q42.1;q14.3) translocation was found to segregate with SZ and other psychiatric disorders, which disrupted a gene termed *DISC1* [50]. Similar to *ErbB4*, *DISC1* is alternatively spliced, producing over 50 distinct isoforms [39]. The expression of three specific isoforms, exon 3-skipping (Δ3), exons 7- and 8-skipping (Δ7Δ8) and exon 3a-inclusion (extra short variant 1 or Esv1) were significantly increased in the hippocampus of SZ patients compared to controls [39]. In addition, the Δ3 mRNA isoform was significantly associated with the SNP rs821597, while the Δ7Δ8 mRNA isoform was significantly associated with rs6675281 and rs821616 [39].

The cellular function of *ErbB4* and *DISC1* in neuronal development has been extensively investigated. *NRG1-ErbB4* signaling has been shown to play critical roles during the development of GABAergic interneurons, axon and dendrite formation, as well as synapse formation [51,52,53]. Loss of *ErbB4* in a mouse model results in decreased GABAergic interneurons [54]. Moreover, *ErbB4* is required to mediate the function of *Nrg1* in stimulating dendrite formation of interneurons [51]. Similarly to *ErbB4*, *DISC1* is highly expressed in the hippocampus [55,56] and enriched in the dentate gyrus of the hippocampus in both mouse models and human postmortem brains, with lower expression in the cerebral cortex and cerebellum [57,58]. Interestingly, the knockdown of *Disc1* in mice resulted in decreased radial migration of neurons within the cortex [59]. These data suggest that the above SZ candidate genes produce multiple alternatively spliced isoforms, play essential functions during development, and their dysregulation in SZ is expected to perturb neuronal circuitry.

The mechanisms that underlie dysregulation of alternative splicing in SZ brains are complex and understudied. While extensive literature has documented the roles of numerous RBPs in controlling alternative splicing, which is essential for advancing neuronal and glial lineage establishment, whether and how RBP abnormalities lead to the dysregulation of alternative splicing in SZ remains elusive. The fact that exome sequencing only uncovered a small percentage of disease-causing genetic alterations suggests that alterations to the non-coding genome may contribute to this debilitating mental disease [60,61,62,63]. Indeed, emerging evidence proposes that lncRNAs may also regulate pre-mRNA alternative splicing besides RBPs and subsequent dysregulation of lncRNAs may contribute to SZ pathogenesis.

## 3. Long Non-Coding RNAs in Schizophrenia

lncRNAs are a large family of non-coding RNAs which harbor greater than 200 nucleotides; however, they do not encode proteins due to the lack of notable open reading frames [64,65]. lncRNAs can be produced from the enhancers, promoters, and introns of coding genes [66,67,68], as well as intergenic bona fide lncRNA genes. In general, lncRNA biogenesis and processing is similar to mRNAs, including transcription by RNA-polymerase II, 5′ capping, canonical splicing to remove introns, and polyadenylation at the 3′ end [69,70,71]. However, some lncRNAs are processed and stabilized in a non-canonical manner. For example, circular RNAs undergo unique back-splicing to form circularized transcripts [72,73,74]. Alternatively, two linear lncRNAs have been shown to form a tRNA-like structure at the 3′ end, which is cleaved by RNase P, resulting in the formation of a 3′ end triple helix structure responsible for stabilizing the transcript [75,76,77,78,79].

Mature lncRNAs often contain multiple sequence domains that are predicted to distinctly bind DNA, RNA, and proteins and hence, function as molecular scaffolds and/or sponges to direct the localization and/or activity of multiple factors simultaneously [80,81,82]. Recent studies revealed the sophisticated roles of lncRNAs in regulating the expression of their downstream target genes [83,84,85,86]. While some lncRNAs are localized to both the nucleus and cytoplasm [87,88], an increasing number of lncRNAs are found restricted to the nucleus [66,89], suggesting they are specialized with distinct mechanisms of action. Some nuclear lncRNAs can act in cis to modulate chromatin and transcription factor binding to the neighboring locus, thus regulating transcription [90,91,92,93]. In addition, *trans-acting* nuclear lncRNAs can organize nuclear subdomains by interacting with various RBPs in distinct nuclear bodies to regulate RNA processing [94,95,96,97,98]. Moreover, cytoplasmic lncRNAs have been reported to sponge RBPs and microRNAs, which, in turn, regulate mRNA translation and stability [99,100,101,102,103,104].

lncRNAs are often expressed in a cell type-specific manner, with over 40% of documented lncRNAs (equivalent to 4000–20,000 lncRNA genes) specifically enriched in the central nervous system [1,105,106,107,108]. This enrichment is substantial given that the GENCODE project revealed that the human genome contains approximately 10,000–50,000 lncRNAs and 20,000–25,000 protein-coding genes [1,106,109]. The multifaceted lncRNA actions can expand molecular and cellular networking, as well as coordination, thus increasing the capacity for greater sophistication of brain function [105,110,111]. Unlike the highly conserved microRNAs, lncRNAs are often poorly conserved, even between human and other mammalian species [112,113,114,115]. Compared to protein-coding genes, lncRNAs are also much less conserved [1,114,116]. Interestingly, brain-enriched lncRNAs exhibit a relatively higher sequence conservation than those in other tissues [117].

Increasing evidence in recent literature indicates an altered expression of lncRNAs in association with neurological and neurodegenerative disorders characterized by cognitive impairment. Recently, autism spectra disorder-associated dysregulation of primate-specific lncRNAs has been reported [118]. In addition, based on an integrated approach combining epigenetic marks, GWAS hits for CNS disorders, and correlation to intellectual disability (ID) genes, several human lncRNAs are potentially involved in ID [119]. Moreover, exon-crossing deletions were found in the large intergenic non-coding (linc) RNA derived from LINC0029 in association with neurodevelopmental abnormalities [120]. Additionally, the monoamine oxidase A (MAOA)-associated lncRNA, called MAALIN, was reported to regulate MAOA expression in impulsive and aggressive suicides, as well as impulsive and aggressive behaviors in mice [121]. Thus, lncRNAs are postulated to underlie intellectual evolution but also cognitive fragility in humans. Given the scaffolding features of lncRNAs and their interaction with multiple types of molecules, lncRNA dysfunction may serve as a converging point to connect various risk factors and underlie the pathogenesis of cognitive illnesses.

Specifically for SZ, fast-growing literature has revealed the association of genetic alterations in lncRNA genes with SZ, aberrant lncRNA expression in SZ patient brains, and lncRNA dysregulation as a potential biomarker for SZ diagnosis and treatment [122,123,124,125,126] (Figure 1). For example, SNPs in the lncRNA gene LINC00461 are significantly associated with SZ subjects showing decreased hippocampal volume. Moreover, this lncRNA was downregulated in the peripheral blood and hippocampus of SZ patients [127]. Another lncRNA implicated in SZ of particular interest in the field is called *GOMAFU*, due to the growing evidence suggesting the function of this lncRNA in normal neuronal development and plasticity, pathophysiology in SZ, as well as its potential to serve as a risk factor and biomarker for SZ.

## 4. Discovery of *GOMAFU* and Disease-Associated Genetic Variants

The mouse *Gomafu* lncRNA was initially described in 2004 as *retinal non-coding RNA 2 (Rncr2)* by a serial analysis of gene expression (SAGE) in the developing retina [128]. This novel mRNA-like spliced transcript of approximately 9 kb with more than 10 alternative isoforms was solely localized in the nucleus and did not encode for any proteins. Moreover, *Rncr2* was reported to play important roles in mouse retinal cell specification [129]. This lncRNA gene was later identified on mouse chromosome 5 and further characterized in an independent study published in 2007 [130], in which genes controlling cell fate specification in distinct mouse retinal cells were identified. At the cellular level, this lncRNA displayed a spotted distribution within the nucleus and was thus named after the Japanese word for spotted pattern, *Gomafu* [130]. *Gomafu* harbors an identical sequence to the 3′ region of EST clones, identified as *RNCR2*, and is abundantly expressed in the brain and subpopulations of retinal neurons. Further studies since this discovery found that although *Gomafu* can interact with well-known splicing factors and is predicted to regulate mRNA splicing [131,132], it is not detected in nuclear speckles where splicing occurs [130,133,134] or paraspeckles that host microRNA biogenesis [130,135]. Rather, *Gomafu* is found in an uncharacterized nuclear domain with unknown composition and function.

The human *GOMAFU* is produced from the *MIAT* gene, which was identified on chromosome 22q12.1 within a susceptibility locus for myocardial infarction (MI) [136]. An early study discovered six SNPs within *GOMAFU* that are significantly associated with MI (Figure 2) [136]. Hence, this human lncRNA was named *myocardial infarction-associated transcript (MIAT)* in 2006, but was later recognized as the human *GOMAFU* orthologue based on moderate sequence homology with mouse *Gomafu* [129,130].

At the time of the initial genetic study, only two of the six SNPs, rs2331291 and rs2301523, were registered in the dbSNP database (build 126), while the remaining four SNPs were denoted by their positions within the lncRNA transcript, namely 8,813 G>A (Exon 3), 11,093 G>A (Exon 5), 11,741 G>A (Exon 5), and 12,311 C>T (Exon 5) (Table 1) [136]. However, with advance of the human genome project, these four SNPs have been registered within the dbSNP (build 155 v2) as rs62224896, rs34403716, rs35955962, and rs35870418, respectively (Table 1). This study provided preliminary genetic evidence for the potential involvement of *GOMAFU* in human disease. Furthermore, this established a foundation for further disease-associated studies for *GOMAFU* through SNP analyses.

While the previous study focused on the SNPs within the *GOMAFU* transcript, another study analyzed whether SNPs near the *GOMAFU* transcript were associated with acute myocardial infarction (AMI) [137]. This study examined the proximal promoter region of *GOMAFU* through PCR amplification using genomic DNA isolated from the peripheral leukocytes of controls and AMI patients in a Chinese Han population. Comparing the SNPs identified from the control and patient group, two SNPs were significantly associated with AMI, rs5752375, and rs9608515 (Table 1 and Figure 2) [137]. These two SNPs are located 938 and 864 bp upstream of the *GOMAFU* transcription start site, respectively, likely within the *GOMAFU* promoter. Moreover, this study proposed transcription factor binding may be altered as a result of the two identified SNPs. However, further analysis into how the polymorphisms influence transcription factor binding and *GOMAFU* expression is needed [137].

While *GOMAFU* has been shown to harbor SNPs linked with myocardial infarction, whether *GOMAFU* harbors SNPs associated with SZ was analyzed only recently [126]. Out of 485 SNPs within *GOMAFU,* eight SNPs were selected based on allelic frequency in a cohort of patients from a Chinese Han population, consisting of 1093 schizophrenic patients and 1180 control patients. Of the eight SNPs analyzed, one SNP, rs1894720, was significantly associated with paranoid schizophrenia (Figure 2) [126]. Interestingly, none of the SNPs associated with myocardial infarction are associated with SZ (Table 1). These findings suggest that SNPs in *GOMAFU* may have different mechanisms involved in MI and SZ vulnerability.

## 5. The lncRNA *GOMAFU* in Neuronal Development and Neurophysiological Function

The enriched expression of *Gomafu* in the brain prompted studies to investigate the role of this lncRNA in normal neuronal development and function, as well as in SZ. Studies showed that in the developing mouse retina, the *Gomafu* lncRNA is expressed in progenitor cells and over the course of development, the expression level of *Gomafu* declines [129]. Depletion of *Gomafu* with shRNA led to an increased production in amacrine cells and Müller glia, suggesting that *Gomafu* may function in regulating retinal cell fate specification. Furthermore, this study also fused distinct *Gomafu* transcript regions to IRES-GFP to be expressed upon transfection into the developing retina. Overexpression of the *Gomafu*-IRES-GFP construct showed that *Gomafu* was translocated to the cytoplasm. The use of additional IRES-GFP constructs encoding for 3′, 5′, and middle regions of the *Gomafu* transcript exhibited distinct functions. Particularly, the 5′ region construct resulted in an increase in amacrine cells, while the 3′ region construct showed an increase in Müller glia. Interestingly, overexpression of the middle region led to an increase in both amacrine cells and Müller glia [129]. These findings suggest *Gomafu* may regulate retinal cell fate specification and this regulation is potentially mediated by specific regions within the *Gomafu* transcript. Whether this property of mouse *Gomafu* is conserved with human *GOMAFU* is not known given the fact that sequences in the spliced mature *GOMAFU* lncRNA are only moderately conserved.

Further exploration into *GOMAFU* function during brain neuronal development came from a study that investigated the role of *Gomafu* during corticogenesis [138]. A combinatorial fluorescent reporter mouse line was generated to isolate proliferative progenitor cells, differentiating progenitor cells, and neurons during corticogenesis. Using in utero electroporation of E13.5 mice, this study revealed that after 2 days of *Gomafu* overexpression, the number of neural progenitor cells in the ventricular zone was increased, which was accompanied by a decrease in the number of differentiating progenitor cells and neurons in the cortical plate. However, RNAi treatment resulted in a similar phenotype as the overexpression studies. The authors propose that *Gomafu* is necessary for neuronal lineage development, yet overexpression of *Gomafu* may increase neuronal death and/or cause more cells to stay in a progenitor state rather than switching to a differentiating state [138]. Moreover, this study provides further in vivo functional evidence for *Gomafu* in regulating neural progenitor cell specification and neuronal survival.

The first evidence for neuronal activity-regulated expression of *Gomafu* appeared in 2014 when Barry et al. showed that the *Gomafu* lncRNA was acutely decreased upon global activation of mouse cortical neurons in culture and human cortical neurons derived from induced pluripotent stem cells (iPSCs) [139]. Moreover, *Gomafu* was reported to be associated with anxiety-like behavior in mice based on paradigms that employ fear conditioning [140]. *Gomafu* was significantly decreased in the medial prefrontal cortex of fear-conditioned mice as compared to control mice. The knockdown of *Gomafu* using antisense oligonucleotides (ASO) within the medial prefrontal cortex did not influence long-term memory in mice but resulted in an increase in anxiety-like behavior, suggesting a role for *Gomafu* in the onset of anxiety disorders [140]. This study further investigated the molecular function of *Gomafu* in regulating gene expression linked with fear-induced anxiety. The intermolecular chaperone, *Crybb1*, which is the mouse homologue of the human SZ risk gene, *CRYBB1*, located in the human 22q12.1 near *MIAT* [141,142], was upregulated upon the knockdown of *Gomafu* [140]. Mechanistically, the possibility for *Gomafu* to regulate proximal gene transcription was further examined [140]. Indeed, RNA immunoprecipitation in cortical neurons revealed interactions between *Gomafu* and the polycomb repressive complex (PRC) component BMI1. Furthermore, the knockdown of *Gomafu* in cortical neurons led to a significant reduction in BMI1 occupancy at the *Crybb1* promoter. These observations suggest that *Gomafu* cooperates with BMI1 to repress the transcription of the nearby *Crybb1* [140].

Recent studies have also explored whether exposure to psychostimulants, such as methamphetamine, influences the expression of lncRNAs *in vivo*, including *Gomafu*. *Gomafu* knockout mice, which did not exhibit global anatomical or physical abnormalities [143], showed hyperactive behavior and increased sensitivity to methamphetamine [143]. An alternative study observed significantly decreased *Gomafu* expression in the prefrontal cortex of mice treated with the NMDA receptor antagonist MK801 or methamphetamine [144]. This result led to the proposed role for *Gomafu* in modulating glutamatergic signaling in the prefrontal cortex [144]. Together with the marked reduction of *Gomafu* upon global neuronal activation in culture [139], these observations suggest that *GOMAFU* is tightly regulated and likely acts as a suppressive modulator to maintain proper neuronal balance. The decline of *Gomafu* may contribute to the exacerbated neuronal activity in response to stress and psychopathological insult. Thus, perturbation of *GOMAFU* function may be related to anxiety and the development of SZ.

## 6. Dysregulation of Human *GOMAFU* Is a Potential Risk Factor for Schizophrenia

In support of the hypothesis that *GOMAFU* malfunction is involved in SZ, Barry and colleagues found a significant downregulation of *GOMAFU* in the superior temporal gyrus in the postmortem cortex of a SZ cohort relative to the controls [139]. Recently, a cross-disorder consortium study by the PsychENCODE Consortium documented transcriptomic profiles in the postmortem frontal and temporal cortex derived from a large patient cohort of SZ, ASD, bipolar disorder (BD), and healthy controls [30,45,145]. In this study, a significantly reduced expression of *GOMAFU* was detected in the brains of SZ and ASD patients, but not BD patients, compared to controls [30]. This observation suggests *GOMAFU* dysregulation is not limited to SZ, but can display certain degrees of disease type specificity.

In an effort to explore the function of *GOMAFU* in human iPSC-derived neurons, *GOMAFU* was found to bind several splicing factors. In addition, the overexpression and ASO-mediated knockdown of *GOMAFU* reciprocally altered the abundance of alternatively spliced variants that are crucial for neuronal development and synaptic function. Moreover, these splice variants represented by the mRNA isoforms of the *v-erb-a erythroblastic leukemia viral oncogene homolog 4 (ErbB4)* neuregulin receptor and the *disrupted in schizophrenia 1 (DISC1)* are dysregulated in SZ [139]. These findings provided the first evidence that *GOMAFU* abnormalities may lead to dysregulated alternative splicing of a SZ risk gene network, which, in turn, contributes to SZ vulnerability and/or pathogenesis.

To expand the potential connection between *GOMAFU* and dysregulated alternative splicing in SZ, another study further investigated the dysregulation of *ErbB4* splicing [146]. Previous work revealed *ErbB4* is mainly expressed in GABAergic interneurons in the adult brain [147,148,149] and parvalbumin interneurons are heavily enriched in the dorsolateral prefrontal cortex layer 4 [150]. Indeed, the SZ-associated *ErbB4* splice variants *Cyt-1* and *Jm-a* are aberrantly increased in SZ parvalbumin interneurons. Interestingly, abnormal upregulation of *GOMAFU* is observed preferentially in parvalbumin interneurons, which is correlated with the dysregulation of *ErbB4* splice variants [146]. These observations support a proposed mechanism by which *GOMAFU* dysregulation may underlie abnormal splicing of SZ risk factor gene transcripts in a neuronal subtype-specific manner. Furthermore, the deficiency of *GOMAFU* in cortical regions and the aberrant increase of *GOMAFU* in parvalbumin interneurons suggests differential dysregulation of *GOMAFU* in distinct neuronal subtypes, which may contribute to the puzzling mechanisms underlying the complex positive and negative clinical phenotypes in SZ (Figure 3A).

## 7. *GOMAFU* in the Peripheral Blood May Serve as a Biomarker for Diagnosis and Treatment for Schizophrenia

The precise diagnosis and treatment efficacy for SZ has been a major challenge. Whether lncRNA abnormalities can offer diagnostic value for brain diseases is an attractive idea to explore. Approximately 80% of brain lncRNAs have been found in peripheral blood mononuclear cells (PBMCs) [125]. Several studies have previously examined whether lncRNA expression in PBMCs may serve as biomarkers for SZ [151,152]. Furthermore, the dysregulation of *GOMAFU* in the postmortem brains of SZ patients raised an intriguing question regarding whether *GOMAFU* expression in PBMCs can serve as a biomarker for the diagnosis and treatment of SZ. To test this idea, *GOMAFU* expression in the PBMCs of SZ patients before and after receiving treatment with antipsychotic medications was examined and compared to healthy controls [152]. Using quantitative reverse transcription polymerase chain reaction, increased levels of *GOMAFU* were detected in PBMCs derived from SZ patients as compared to healthy controls [153]. Moreover, treatment with antipsychotic drugs for a 12-week period further increased the expression of *GOMAFU* in PBMCs as compared to the SZ patients before treatment. This study provided evidence that *GOMAFU* expression in the blood is correlated with the diagnosis and treatment of SZ. However, the authors acknowledged several limiting factors for this study, including the small sample size of patients recruited to the study and the unsolved issue of whether the change in *GOMAFU* expression reflects improvement by the treatments or progression of the disease [153].

In a continued effort to explore the potential use of lncRNAs as biomarkers for SZ, a recent study investigated nine lncRNAs, including *GOMAFU*, to explore their association with SZ patients from a Chinese Han population before and after antipsychotic treatment [125]. *GOMAFU* was associated with and predictive of SZ pathogenesis individually and collectively together with other lncRNAs. Noticeably, *GOMAFU* expression was significantly upregulated in the plasma of SZ patients compared to controls, while the majority of other lncRNAs were downregulated [125]. Additionally, in a distinct SZ cohort, *GOMAFU* expression in the peripheral blood was analyzed in parallel with three other lncRNAs and was shown to discriminate SZ patients from normal subjects with a diagnostic power of 68% [154]. These newer findings provide additional support for the usage of lncRNAs, including *GOMAFU*, as potential biomarkers for the diagnosis of and prognosis for SZ. Although *GOMAFU* dysregulation was also identified in the postmortem cortical areas of ASD brains [30], *GOMAFU* expression in the peripheral blood is only examined and reported in SZ subjects and this has not been examined in ASD patients. Thus, whether *GOMAFU* abundance in the PBMCs can also be used as a biomarker for other brain diseases needs to be further investigated.

## 8. Mechanisms for *GOMAFU* to Regulate the Downstream Molecular Pathways in Schizophrenia

*GOMAFU* dysregulation in SZ has attracted increasing attention to this lncRNA and prompted investigations into the mechanisms of *GOMAFU* function through interactions with RBPs (Figure 3B). Sone et al. investigated the localization and function of *Gomafu* in a subset of postmitotic neurons and found that *Gomafu* predominantly localizes within the nucleus based on studies that employed in situ hybridization [130]. Furthermore, they observed a distinct intranuclear localization pattern of *Gomafu,* which did not associate with chromatin or colocalize with markers for paraspeckles, speckles, Cajal bodies, and this suggested *Gomafu* marks a novel nuclear domain [97,130]. The subnuclear *GOMAFU*−RBP interactome and mechanisms that underlie the nuclear distribution pattern of *GOMAFU* have not yet been determined. Interestingly, a tandem repeat of UACUAAC in the *GOMAFU* RNA transcript was identified, which is a known interacting sequence motif for splicing factor 1 (SF1) [131]. Expansion of this tandem repeat does not influence the nuclear localization of *GOMAFU*. Furthermore, *GOMAFU* competitively binds to SF1 with a high affinity to influence the splicing kinetics of pre-mRNAs. Interestingly, while SF1 is distributed throughout the nucleus, colocalization with *GOMAFU* only occurs in distinct spots [131]. These observations support the role of *GOMAFU* in regulating splicing, though the precise mechanism remains elusive.

Since *Gomafu* is primarily localized in the nucleus and interacts with RBPs and splicing factors, a study explored the potential for *Gomafu* to interact and sequester two splicing RBPs, Celf3, and SF1 [132]. The authors explored the expression and localization of Celf3 and SF1 and determined that these RBPs form specialized nuclear structures, which they termed CS bodies in the absence of *Gomafu* in Neuro2a cells. The authors proposed that *Gomafu* sequesters these RBPs from the CS bodies into distinct nuclear domains [132]. On the other hand, the knockdown of Celf3 led to significant downregulation of *Gomafu*, indicating mutual regulation between *Gomafu* and the RBPs it interacts with. In addition, *Gomafu* was shown to interact with other splicing factors, including SRSF1 and quaking (QKI) [139]. Together, these findings suggest that during brain development, *GOMAFU* may modulate the functional availability of various RBPs to regulate the homeostasis of a broad RNA network.

The potential interaction of *GOMAFU* with QKI is an interesting finding given that *QKI* has been identified as another candidate gene for SZ. The *quaking (qkI)* gene was initially discovered proximal to a deletion site in a mouse model, known as *quaking viable (qk^v^)* where an autosomal recessive mutation resulted in severe tremors and dysmyelination [155,156,157]. These mice also show cell-type specific alterations in the splicing of the *qkI* gene [155]. Moreover, an additional *qkI* mutation in the mouse, *qk^kt4^*, has been shown to be embryonic lethal, and the translated protein product eliminates the dimerization of the QKI protein [158,159]. Three protein products with distinct C-termini (QKI-5, -6, and -7) are produced through alternative splicing of the 3′ exons within the QKI pre-mRNA [41,159,160]. QKI-5 is predominantly localized in the nucleus, whereas QKI-6 and QKI-7 are localized in the cytoplasm [161,162]. QKI isoforms are initially expressed in neural stem cells (NSCs) in the brain but restricted in glia lineage to advance glial cell differentiation during later development [163,164]. All QKI isoform proteins bind to the quaking response element (QRE) consensus sequence on mRNA targets to regulate mRNA splicing, stability, and translation [165,166,167,168]. The nuclear QKI-5 protein isoform has been shown to function as a regulator of the alternative splicing of numerous transcripts in NSCs during embryonic development and later on, in oligodendroglia in which QKI is most abundantly expressed, including myelin basic protein and myelin-associated glycoprotein, and thus modulates myelination [41,166].

Several studies have elucidated QKI mRNA steady state levels are dysregulated in various regions of SZ brains as well as potential connections to SZ etiology [29,40,169,170]. In a pedigree of a Swedish family, chromosome 6q25-6q27 contains a susceptibility locus for SZ [40,171,172] and strikingly, within a specific region identified by SNP markers and microsatellites, the only gene present is *QKI,* which is located on chromosome 6q26 [40,173]. An analysis of QKI isoform mRNA expression revealed QKI-7 and QKI-7b mRNA expression was significantly decreased in SZ patients compared to controls [40]. Moreover, all QKI isoforms exhibited increased mRNA expression in patients treated with typical antipsychotic medications compared to controls [40]. In addition, the characterization of myelination marker genes revealed decreased steady state mRNA expression in SZ patients compared to controls [40]. Several of the marker genes analyzed have previously been proposed to be regulated by QKI [41,163,169,170], suggesting the dysregulation of QKI in SZ influences the alternative splicing of myelination genes. An additional independent study analyzing the anterior cingulate gyrus reported the steady state levels of QKI-5, -6, and -7, as well as pan-QKI were significantly decreased in SZ patients compared to controls [29]. These studies provide increased support for how pre-mRNA alternative splicing is dysregulated in SZ, further elucidating the potential to harness such dysregulation for identifying additional regulatory and diagnostic mechanisms. Considering the essential roles of QKI in neuronal and glial lineage development, the reported genetic variants in the QKI gene associated with SZ [40], and the dysregulation of QKI in SZ postmortem brains [29,139], QKI and *GOMAFU* may form a potentially intricate risk factor network for SZ that affects splicing, most likely in NSCs and early neuronal linage development when both exhibit key roles.

## 9. Conclusions

Decades of efforts have been made to characterize the expression and function of lncRNAs [3,6,7,83,84]. lncRNAs play multifaceted roles in regulating nuclear organization, transcription, splicing, and translation [81,85,88,90,91,94,99]. Furthermore, accumulating evidence conceptually advanced the importance of lncRNAs in governing the differentiation of neural cells, regulation of neuronal plasticity, and lncRNA abnormalities in SZ [42,111,174,175,176,177,178]. Specifically, the neuronal lncRNA *GOMAFU* has attracted increasing attention due to genetic alterations and the functional involvement of this lncRNA in relation to neuronal development, neurophysiology, and SZ. Recent investigations into the usage of *GOMAFU* as a biomarker for SZ have further piqued the interest in *GOMAFU*. Despite these exciting discoveries, limitations and prevailing knowledge gaps clearly exist.

One common limitation of genetic linkage analysis, biomarker identification, and postmortem studies is the cohort sample size. In addition, the environmental factor influence may also affect the expression level of *GOMAFU,* but, unfortunately, this has not been factored into the conclusion. Specifically for biomarker studies, whether and how expression levels in the peripheral blood indeed translate to expression in the brain needs to be further addressed. Even within the brain, whether *GOMAFU* expression is differentially affected in a brain region-specific manner in various psychiatric diseases has not been determined. Moreover, whether distinct neuronal subtypes employ different molecular mechanisms to regulate *GOMAFU* and how such regulation is disrupted in SZ have not been explored.

Another major challenge is elucidating the molecular mechanisms that underlie the multifaceted function of lncRNAs. While multiple lines of evidence suggest a role for *GOMAFU* in organizing subnuclear domain distribution of RBPs and regulating alternative splicing of SZ risk factor gene transcripts, the number of studies investigating lncRNA−RBP interactions in relation to SZ is very limited [131,132,136,139,146]. Recent advancements in method development have led to several new ways to identify lncRNA−RBP interactomes. Comprehensive identification of RNA-binding proteins (ChIRP) and capture hybridization analysis of RNA targets (CHART) are two newly developed methods for in silico identification of lncRNA-interacting RBPs without prior knowledge of the lncRNA-associated RBPs [179,180,181,182,183,184] (Figure 3C). These two methods provide a new avenue for studies to not only reveal novel RBPs that interact with lncRNAs, but also provide a method for determining how alterations in lncRNAs can influence RBP localization and activity. Specifically, identification of the *GOMAFU*−RBP interactome may provide critical clues into how *GOMAFU* dysregulation may affect RBP networks and the dynamic landscape of alternative splicing, which is a crucial factor in normal neuronal development and the pathogenesis of SZ.

## Figures and Tables

**Figure 1 cells-11-01949-f001:**
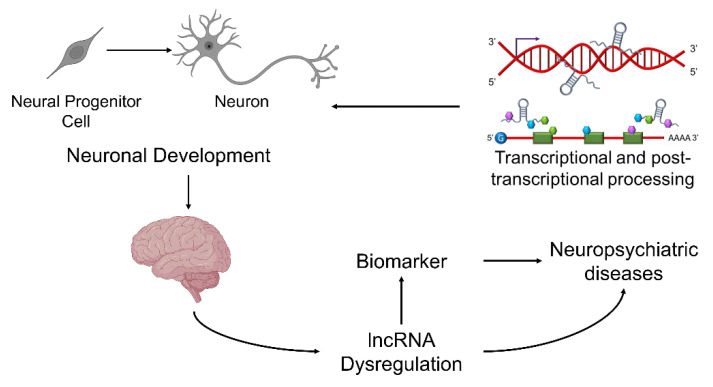
Nuclear lncRNAs function to regulate transcriptional and post-transcriptional processing which governs the development and function of neurons. Dysregulation of lncRNA in the central nervous system, specifically in neurons, is thought to contribute to the pathogenesis of SZ and may serve as a biomarker for diagnosis and treatment.

**Figure 2 cells-11-01949-f002:**
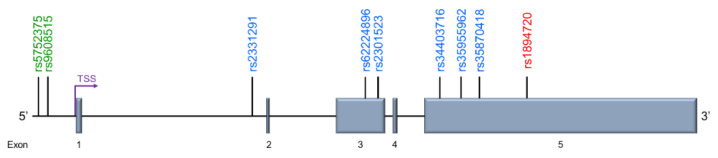
Schematic representation of SNPs in the *GOMAFU* locus that are associated with acute myocardial infarction (Green), myocardial infarction (Blue), and schizophrenia (Red). The transcription start site (TSS) is indicated by an arrow.

**Figure 3 cells-11-01949-f003:**
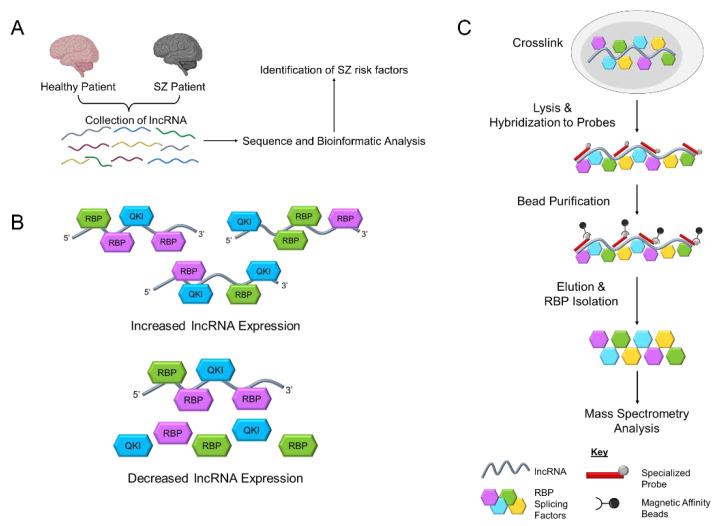
lncRNAs have diverse cellular functions in SZ. (**A**) Comparing transcriptomic alterations of lncRNAs, including *GOMAFU*, between healthy controls and SZ patients provides insights into how lncRNAs are potential risk factors for SZ. (**B**) lncRNA are capable of interacting with and modulating the function of RNA-binding proteins (RBPs). Increased expression of *GOMAFU* results in sequestration of RBPs, including the schizophrenic-related RBP quaking (QKI). In contrast, decreased *GOMAFU* expression releases RBPs to regulate their targets. (**C**) New methods have been developed to characterize the interactions between lncRNAs and RBPs. In general, these new methods involve crosslinking lncRNA with RBPs in vivo, lysing cells, and hybridizing probes to lncRNA of interest. This is followed by purification using magnetic beads, elution of specific complexes, and isolation of RBPs. Novel RBPs in the lncRNA−RBP interactome are then identified by mass spectrometry and further confirmed by immunoblotting. Colored hexagons are representative RBPs.

**Table 1 cells-11-01949-t001:** Acute myocardial infarction, myocardial infarction, and paranoid schizophrenia SNPs significantly associated with the lncRNA *GOMAFU* and their chromosomal location.

Disease	dbSNP	Alternative Identification	SNP Position	Chromosomal Location	Allele	Ref.
Acute Myocardial Infarction	rs5752375	g.4063 T>C	−938	26656544	T/C	[137]
	rs9608515	g.4137 T>C	−864	26656618	T/C	
Myocardial Infarction	rs2331291	-	5376	26662857	C/T	[136]
	rs62224896	Exon 3 8813	8851	26666332	G/A	
	rs2301523	-	9224	26666705	G/A	
	rs34403716	Exon 5 11,093	11,132	26668613	G/A	
	rs35955962	Exon 5 11,741	11,780	26669261	G/A	
	rs35870418	Exon 5 12,311	12,350	26669831	C/T	
Paranoid Schizophrenia	rs1894720	-	13,780	26671261	G/T	[126]

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
