# Peer review of "The Long Non-Coding RNA GOMAFU in Schizophrenia: Function, Disease Risk, and Beyond"

_cells, 2022, doi:10.3390/cells11121949_

Round 1

Reviewer 1 Report

This is a nicely written review, aiming at addressing the role of the lncRNA GOMAFU in neuropsychiatric diseases with focus on schizophrenia.

The review is partly informative and includes helpful figures and tables, but stays partly imprecise and superficial for different aspects.

However, some improvements are required before acceptance.

In the title, the role of GOMAFU in neuropsychiatric diseases is claimed. However, neither these diseases are comprehensively introduced nor I see the focus on neuropsychiatric diseases but rather on schizophrenia. So, the title should be more adapted to the content.

The second paragraph (Long Non-coding RNAs in Neuropsychiatric Diseases) should contain a broader intro about neuropsychiatric diseases (if keeping the focus on more than schizophrenia) and also more intro about schizophrenia. Further, lncRNAs could be introduced in an extra chapter.

When introducing schizophrenia, the causes and the predisposition genes such as ERRB4 and DISC1 should be mentioned, also the putative role of developmental defects also in matters of GABAergic interneurons and the regulation of their proper development by predisposition genes such as ERRB4 and DISC1 should be outlined. The intro into SZ comes too short. If focussing on more diseases, they should also be properly introduced.

I liked very much the connection of lncRNA function in intellectual evolution and cognitive fragility in humans. It would be favourable, if this would be more elaborated.

In line 306, the authors write: „Furthermore, the dysregulation of GOMAFU in postmortem brains of SZ and ASD patients raised an intriguing question regarding whether GOMAFU expression in PBMCs can serve as a biomarker for neuropsychiatric diseases.“ However, GOMAFU was found altered in the brains of SZ and ASD but not BD patients. How can they then be a biomarker for neuropsychiatric diseases? If the authors introduce more ASD, BD, and SZ (symptoms, disease pathophysiologies, predisposition genes etc.), there might be revealed a common link between GOMAFU function and symptoms in ASD and SZ (e.g. cortical interneurons???), which would be very interesting. At least, the authors need to be more precise here. Biomarker for what exactly? Finally, only examples for SZ are provided.

Moreover, in paragraph 7, the potential mechanistic of QKI as GOMAFU-interacting SF  for schizophrenia disease pathophysiology could be more elaborated, but shedding more light on the QKI role and splicing in general in neuronal development. Alternative splicing and SZ risk could be also more in detail highlighted in the Intro-part of SZ or neuropsychiatric diseases.

Minor issues:

Figure 1 contains „frames“, please correct.

line 171: GOMAFU (U is missing)

Reviewer 2 Report

The authors provide an interesting review looking into the potential role of GOMAFU in neuropsychiatric disorders. The authors summarize the relevance of GOMAFU-related molecular pathways in SZ. The study provide more evidence for understanding the mechanism of SZ.

Specific comments to be addressed are:

1. Most part of the abstract is related to introducing lncRNAs, with very few contents to describe their own results. A balanced abstract is needed.

2. Some readers may not clearly know why the authors select this particular lncRNA. So the authors need to mention other lncRNAs as a backgroud introduction, like LINC00461.

Reviewer 3 Report

This is a timely, comprehensive, and critical review of a fragment of nascent, intriguing and important field of the role of long non-coding RNAs in neuropsychiatric diseases. The review is well-organized and helpfully illustrated. 

Author Response

Thanks for the appreciation of our work. We carefully checked English language and style as suggested.

Round 2

Reviewer 1 Report

The revised manuscript is now much more focussed and provides sufficient depth, representing an added value to the scientific literature.

Apart from minor spelling mistakes (e.g.: line 449: may modulate) that should be corrected throughout the manuscript, I still have one aspect.

In the initial review, I critzised the following that was answered by the authors also as follows:

"6. In line 306, the authors write: “Furthermore, the dysregulation of GOMAFU in postmortem brains of SZ and ASD patients raised an intriguing question regarding whether GOMAFU expression in PBMCs can serve as a biomarker for neuropsychiatric diseases.” However, GOMAFU was found altered in the brains of SZ and ASD but not BD patients. How can they then be a biomarker for neuropsychiatric diseases? If the authors introduce more ASD, BD, and SZ (symptoms, disease pathophysiologies, predisposition genes etc.), there might be revealed a common link between GOMAFU function and symptoms in ASD and SZ (e.g. cortical interneurons???), which would be very interesting. At least, the authors need to be more precise here. Biomarker for what exactly? Finally only examples of SZ are provided. 

We accept this criticism and agree that the evidence in the literature indicates GOMAFU may be a biomarker specific for SZ. We have altered the text of this section and specifically discuss the idea of GOMAFU expression levels in the blood may be a biomarker for the diagnosis and treatment of SZ. "

In the new version, the authors now deleted AD from their sentence (line 394 ff). However, as obviously dysregulation of GOMAFU was also seen in AD brains postportem (as reported previously), it brings me back to my initial question: How can this then be a specific biomarker for SZ? Just deleting AD will not help here. Is there any aspect that makes GOMAFU a potential marker of SZ but not AD, albeit being obviously dysregulated in both diseases?

Author Response

Thank you for the review and instruction to further clarify the potential of GOMAFU as biomarker for SZ.

The reviewer indicated: "In the new version, the authors now deleted AD from their sentence (line 394 ff). However, as obviously dysregulation of GOMAFU was also seen in AD brains postportem (as reported previously), it brings me back to my initial question: How can this then be a specific biomarker for SZ? Just deleting AD will not help here. Is there any aspect that makes GOMAFU a potential marker of SZ but not AD, albeit being obviously dysregulated in both diseases?"

We revised the text to add back the deficiency of GOMAFU in the ASD brains, while indicating that “Although GOMAFU dysregulation was also found in the postmortem cortical areas of the ASD brains, GOMAFU expression in the peripheral blood is only examined and reported in SZ subjects, but has not been examined in ASD patients. Thus, whether GOMAFU abundance in PBMCs can also be used as a biomarker for other brain diseases still waits for future investigation."

We also carefully checked and corrected minor errors in spelling as suggested.